# Analysis of Procollagen C-Proteinase Enhancer-1/Glycosaminoglycan Binding Sites and of the Potential Role of Calcium Ions in the Interaction

**DOI:** 10.3390/ijms20205021

**Published:** 2019-10-10

**Authors:** Jan Potthoff, Krzysztof K. Bojarski, Gergely Kohut, Agnieszka G. Lipska, Adam Liwo, Efrat Kessler, Sylvie Ricard-Blum, Sergey A. Samsonov

**Affiliations:** 1Faculty of Chemistry, University of Gdańsk, ul. Wita Stwosza 63, 80-308 Gdańsk, Poland; 2Institute of Chemistry and Biochemistry, Free University of Berlin, Takustr. 3, 14195 Berlin, Germany; 3Institute of Materials and Environmental Chemistry, Research Centre for Natural Sciences, Hungarian Academy of Sciences, Magyar tudósok körútja 2, 1117 Budapest, Hungary; 4MTA-ELTE Research Group of Peptide Chemistry, Hungarian Academy of Sciences, Eötvös Loránd University, Budapest 112, P.O. Box 32, 1518 Budapest, Hungary; 5Maurice and Gabriela Goldschleger Eye Research Institute, Tel-Aviv University Sackler Faculty of Medicine, Sheba Medical Center, Tel-Hashomer, 52621 Ramat Gan, Israel; 6Institute of Molecular and Supramolecular Chemistry and Biochemistry, UMR 5246, CNRS, INSA Lyon, CPE, University Claude Bernard Lyon 1, Univ Lyon, 69622 Villeurbanne CEDEX, France

**Keywords:** procollagen C-proteinase enhancer-1, glycosaminoglycans, computational analysis of protein-glycosaminoglycan interactions, calcium ions, fragment-based docking

## Abstract

In this study, we characterize the interactions between the extracellular matrix protein, procollagen C-proteinase enhancer-1 (PCPE-1), and glycosaminoglycans (GAGs), which are linear anionic periodic polysaccharides. We applied molecular modeling approaches to build a structural model of full-length PCPE-1, which is not experimentally available, to predict GAG binding poses for various GAG lengths, types and sulfation patterns, and to determine the effect of calcium ions on the binding. The computational data are analyzed and discussed in the context of the experimental results previously obtained using surface plasmon resonance binding assays. We also provide experimental data on PCPE-1/GAG interactions obtained using inhibition assays with GAG oligosaccharides ranging from disaccharides to octadecasaccharides. Our results predict the localization of GAG-binding sites at the amino acid residue level onto PCPE-1 and is the first attempt to describe the effects of ions on protein-GAG binding using modeling approaches. In addition, this study allows us to get deeper insights into the in silico methodology challenges and limitations when applied to GAG-protein interactions.

## 1. Introduction

Glycosaminoglycans (GAGs) are anionic periodic linear polysaccharides, which are composed of periodic disaccharide units [1] and play a key role in many biologically relevant processes by interacting with their numerous and diverse protein targets such as cytokines and growth factors in the extracellular matrix [2,3,4,5]. However, the molecular mechanisms underlying GAG-mediated interactions are not fully understood, and experimental techniques alone are not sufficient for gaining insights into them [6]. Molecular modeling approaches are not only complementary to experiments, but also provide additional and crucial details, which are experimentally inaccessible. In our previous work, we successfully applied molecular docking and molecular dynamics methodologies in order to model protein-GAG interactions. In particular, we have modeled the effects of GAG binding on chemokines [7,8], growth factors [9,10] and other proteins [11,12], which allowed us to investigate the fundamental questions related to these interactions such as their specificity, the role of multipose character of GAG binding and polarity of binding poses of these periodic molecules.

In this work, we model interactions of GAGs with procollagen C-proteinase enhancer-1 (PCPE-1, encoded by gene *PCOLCE*), a glycoprotein which plays an important role in the assembly of the extracellular matrix [13,14]. Lacking proteolytic activity on its own, PCPE-1 enhances C-terminal procollagen processing, mediated by tolloid-like proteinases such as bone morphogenetic protein 1 (BMP-1) and mammalian tolloid (mTLD) designated BMP-1/tolloid-like proteinases (BTPs) [14,15,16,17]. PCPE-1 expression is upregulated in fibrosis [18,19]. PCPE-1 comprises two complement, sea urchin protein Uegf, BMP-1 (CUB) domains [20] and a netrin-like (NTR) domain [21]. Although neither an X-ray nor an NMR structure is available for full-length PCPE-1, X-ray structure of CUB1-CUB2 domains (PDB ID: 6FZV, 2.7 Å) in a complex with C-propeptide of procollagen [22] and NMR structure of the NTR domain (PDB ID: 1UAP) are available [23]. In the structure of the active CUB1-CUB2 fragment of PCPE-1 bound to the C-propeptide trimer of procollagen III (CPIII), two Ca^2+^ ions participate in the formation of the interface between the CUB1-CUB2 domains and the procollagen III molecule [22]. Often, CUB domains bind Ca^2+^, and Ca^2+^ coordination involves acidic amino acid residues (i.e., Tyr-Glu-Asp-Asp motif) [24]. A conserved calcium binding site has indeed been identified in the CUB1 domain of PCPE-1, and mutational analysis of this site confirmed that PCPE-1 stimulating activity requires a calcium binding motif in the CUB1 domain, which is highly conserved among CUB-containing proteins [25]. A low-resolution structure of the full-length PCPE-1 protein was proposed based on small angle X-ray scattering (SAXS), analytical ultracentrifugation and transmission electron microscopy, showing that PCPE-1 is a rod-like molecule, with a length of 150 Å [26]. PCPE-1 binds to heparin (HP) as shown using affinity chromatography [27] and surface plasmon resonance (SPR) binding assays [28], and the binding is mediated by the NTR domain. Heparan sulfate (HS) and dermatan sulfate (DS) but not chondroitin sulfate (CS) inhibit PCPE-1-HP interactions. HP also binds to BMP-1 [29]. HS could thus potentially act as a scaffold to assemble BMP-1, PCPE-1 and procollagen together at the cell surface [28]. Therefore, the characterization of PCPE-1/GAG interactions at the atom level is important for the detailed understanding of PCPE-1 functions. 

The aim of this work is to get deeper insights into PCPE-1/GAG interactions using both SPR inhibition assays and in silico techniques to complement the experimental data obtained in the previous [28] and present work. Modeling approaches were used to build structural models of full-length PCPE-1 and to determine GAG specific binding to PCPE-1 and its domains. We analyzed the binding of PCPE-1 to GAGs of different types, lengths and sulfation patterns, which were rationally and systematically chosen to match those used in experiments. We also investigate the potential role of Ca^2+^ in these interactions [28] and evaluate the challenges of in silico methodology to study protein-GAG interactions [30]. The results reported here contribute to the understanding of the biologically relevant PCPE-1/GAG interaction and, for the first time, systematically predict the structural positions and the effects of Ca^2+^ ions on protein-GAG complexes. 

## 2. Results and Discussion

### 2.1. Experimental Results

We have previously shown that DS, HS and HP but not CS inhibited the binding of soluble PCPE-1 to immobilized HP [28]. Here, we investigated the effect of HP oligosaccharides of various length as inhibitors of PCPE-1 binding to HP in order to determine the optimal size of HP required to bind to PCPE-1. There was a trend towards an increase in inhibition of PCPE-1-HP interaction with the length of HP oligosaccharides from dp2 to dp8, and then from dp14 to full-length HP chains (Figure 1). HP decasaccharides and dodecasaccharides (dp10 and dp12, respectively) inhibited the binding of PCPE-1 to HP to a lesser extent than the HP octasaccharide (dp8). The oligosaccharides used for inhibition experiments were separated according to their size and not to their sulfation pattern and/or charges. They thus contain a mixture of oligosaccharides of the same size displaying a different number of sulfate groups in different positions of their sequences resulting in different binding motifs with likely different inhibitory efficiencies. This heterogeneity might be more pronounced in dp10 and dp12, leading to a lower global inhibition by these oligosaccharides than by the octasaccharide. 

Then we applied the in silico approaches we have previously developed to analyze the binding of PCPE-1 to GAGs at the atomic level and to determine if these interactions were exclusively electrostatic-driven or if other factors modulate the binding strength.

### 2.2. Modeling the Full Structure of PCPE-1

We created two ensembles of full-length PCPE-1 structures using the UNRES (from UNited RESidue) coarse-grained (CG) approach to determine the structure of the linker located between the CUB1-CUB2 and the NTR domains. In the first one, the structures of the linkers were optimized, and the domain structures were restrained, while in the second one, SAXS derived restraints were used additionally in order to reproduce the experimental data [26] (see Section 3.4 for more details). Five most probable structural models were obtained for both ensembles. For HP binding analysis we used the first three models obtained without SAXS restraints and one model obtained with SAXS restraints (SAXS Model) (Table 1). The radii of gyration of the models obtained without SAXS restraints were significantly lower than those of the elongated structures restrained using SAXS data. As expected, the SAXS Model had a radius of gyration in agreement with the experimental value calculated using SAXS (41 ± 3 Å versus 43 ± 1 Å [26]. The obtained model was also consistent with the length of the protein determined experimentally (150 Å). Poisson‒Boltzmann surface area (PBSA) calculations applied to these 4 models suggest that potential binding regions for HP were located in the NTR domain for Model 3, at the interface of the linker and the NTR domain for Model 2 and SAXS Model, and at the common interface of all domains (CUB1-CUB2, linker and NTR) in Model 1 (Figure 2, Appendix A). 

### 2.3. PCPE-1 Interactions with Glycosaminoglycans

We modeled and analyzed the binding of PCPE-1 and its domains, NTR and CUB1-CUB2, with the following GAGs: chondroitin sulfate-6 (CS6) made of two GalNAc6S-GlcA or three disaccharide units (dp4 and dp6, respectively), dermatan sulfate comprised of three GalNAc6S-IdoA disaccharide units (dp6), and heparin (HP) made of one, two and three GlcNS6S-IdoA2S disaccharide units (dp2, dp4, and dp6 respectively). These GAGs were selected for the following reasons: to compare the in-silico data with the experimental ones previously obtained with these GAGs [28] and to investigate the effects of epimerization, length and sulfation pattern of GAGs on binding. Conventional docking approaches are severely limited in terms of the size of GAGs and can be effectively used only for the GAGs with a length up to dp6 [31]. Therefore, we used HP oligosaccharides of different lengths, from dp2 to dp6, to determine the effect of the GAG length on the binding to PCPE-1. Since HP is the strongest binder, the results obtained with HP oligosaccharides of different lengths should be the most representative. Furthermore, the GAGs studied here were selected in order to systematically evaluate the changes in binding to PCPE-1 according to the GAG length (dp4‒dp6 for CS6 and dp2‒dp6 for HP), the epimerization of glucuronic acid (CS6 dp6 and DS dp6), the increase in the number and position of sulfated groups (i.e., the sulfation pattern) and the net charge of the oligosaccharides (CS6, DS and HP). 

Several clusters of docking solutions were obtained for each GAG tested. The polarity of the binding poses was analyzed because the orientation of the GAG chain was shown to be non-random for the IL-8 chemokine [7] and determinant for the binding specificity of the C-X-C motif chemokine ligand 14 [8], suggesting an important functional role of GAG polarity in their interactions with proteins. Then, for the most diverse binding poses within these clusters, molecular dynamics (MD) simulations were performed with binding free energy post-processing calculations and per residue binding free energy decomposition. We would like to emphasize that choosing a proper procedure of pose selection for such analysis is very challenging, since it is unclear how many clusters and solutions within each cluster should be representative, which part of the trajectory should be analyzed in terms of the free energy, if only the best scored pose from a cluster or all the poses should be taken into account for the further calculations, and how to weight their contributions in the latter case. The answers to these questions are dependent on the molecular systems and on the particular goal of the modeling study. These methodology-oriented aspects of protein-GAG modeling will be further discussed below. 

#### 2.3.1. The NTR-Domain

Among the found clusters of docking solutions, for CS6 dp4 and HP dp2, dp4 and dp6, one major cluster was observed, while there was more uniform distribution of the solutions between several clusters for CS6 dp6 and DS dp6 (Table 2). This suggests that for those molecules, especially for CS6 where the clusters are especially diverse, multipose binding might be quite probable. GAG multipose binding was previously identified both experimentally and computationally for TIMP-3, which is homologous with the NTR domain [11]. Most solutions were localized near the C-terminal α-helix of the NTR domain except for CS6 dp6 (Figure 3 and Figure 4). The size of the clusters obtained by molecular docking was not correlated with their corresponding free binding energies calculated from the MD simulation. This means that molecular docking alone was not able to properly score the solutions, although the Autodock 3 (AD3) scoring function is one of the most successful scoring schemes when applied to GAG complexes [10]. Similarities of the binding regions for the docking solutions post-processed by MD-based binding free energy decomposition per residue are reflected in Table 3 and Table 4 for the obtained clusters and for each GAG ligand respectively. According to the binding free energy values obtained for the NTR domain bound to GAGs compared to the experimental complexes from the PDB [31] and given that no dissociation of these complexes was observed, we assume that the binding of the analyzed GAGs to NTR is stable. The binding strength, evaluated by the calculation of free binding energy, of CS6 dp4 and CS6 dp6 did not significantly differ, but the cluster location of CS6 dp6 differed from those of CS dp4, DS dp6 and HP dp2, dp4 and dp6. Only the third biggest cluster for CS6 dp6 was located in the region overlapping with those of other analyzed GAGs. CS6 dp6 and longer CS6 oligosaccharides might thus bind NTR differently from CS6 dp4 and other GAGs. Therefore, although the binding strength was similar for CS6 and DS, their preferred binding sites were distinct for these two GAGs, which differ only in the epimerization of glucuronic acid. This could potentially explain the results from surface plasmon resonance binding assays, which showed that CS6 did not inhibit PCPE-1 binding to HP whereas DS did [28]. Whereas DS competes with HP for the same binding site on PCPE-1, CS6 binds to a different region, which would allow HP oligosaccharides to remain bound to the NTR domain. Similar computational approaches were successfully applied to demonstrate the experimentally proven differences in binding strength between DS and CS6 interacting via the same binding pose to IL-8 [7,32] In contrast, the binding differences for those GAGs were related to certain differences in the binding pose for CXCL14 [8]. This suggests that for protein-GAG complexes, the predictive power of the computational methods is dramatically dependent on the protein involved and the distribution of the clusters on its surface, which is, in turn, also sensitive to a particular clustering procedure. HP binds the NTR domain stronger than CS6 and DS, while its increase in length stabilizes the interaction suggesting a key role of electrostatic interactions, although few hydrophobic amino acid residues (leucine and valine) and polar, uncharged, amino acid residues (asparagine and glutamine) were predicted to interact with the analyzed GAGs (Figure 4). Most clusters revealed a bias towards specific polarity of GAG binding poses, although this trend was less pronounced for HP dp6 and DS dp6 (Table 2). This suggests that our docking approach is able to distinguish GAG polarity, which is an important methodological finding and will allow us to investigate one of the potential parameters underlying the specificity of protein-GAG interactions [8].

#### 2.3.2. CUB1-CUB2 Domains

Although there is no experimental evidence suggesting that CUB1-CUB2 domains of PCPE-1 directly interact with GAGs, the differences in binding of NTR and full-length PCPE-1 to HP and HS [28] indicate that CUB1-CUB2 domains could affect GAG binding to the full-length PCPE-1 protein. Therefore, we analyzed the potential binding of these domains to GAGs using the same procedure as above.

All the predicted binding poses were either located in the cleft region between the CUB domains or bridged both CUB domains (Appendix A). In both cases, such potential binding would lead to restricted movements of the CUB domains relative to each other, which, in turn, would affect the overall flexibility of PCPE-1 and its ability to recognize and to bind its partners. Calculated GAG free binding energies were essentially less favorable than those calculated for the NTR domain (Appendix A), which is consistent with NTR being responsible for GAG binding in PCPE-1. No binding poses of the analyzed GAGs or the structures that can be obtained from them by GAG chain elongation were found to be in close proximity to the Ca^2+^ binding sites or at the interface with procollagen peptides [23]. In a number of cases, the binding poses predicted by molecular docking were unstable (ΔG higher than −15 kcal/mol), and the GAG dissociated from the protein. Such behavior was typically observed for HP oligosaccharides and is explained by the repulsion of these highly charged molecules by the negatively charged residues of the CUB1-CUB2 domain. Poses corresponding to the binding of CS6 and DS, which are less negatively charged than HP, were globally more stable. However, some binding poses were very stable and comparable with those found in the NTR domain (e.g., cluster 2, solution 2 for CS6 dp4). In such cases, bound GAGs protruded deeply into the cleft between the CUB1 and CUB2 domains forming strong van der Waals interactions in addition to the electrostatic interactions, which are believed to be the driving force in the formation of protein-GAG complexes [31,34]. As reported for the NTR domain, highly significant differences in free energy were found for GAGs within and beyond the same clusters. One major cluster was found for CS6 of various length in contrast to what was observed for other GAG analyzed. No correlation was found between the size of clusters and their free binding energies. The comparison between the observed clusters and the data averaged for different GAGs in terms of the most important protein binding residues showed high similarities for all GAGs, suggesting weak and rather unspecific binding to CUB1-CUB2 domains (Appendix A, Appendix A). Interestingly, for CS6 dp4 the differences between the clusters were more prominent than the differences of these clusters with those obtained for other GAGs. The increase in length of HP from dp2 to dp6 did not modify the potential interaction pattern with the CUB1-CUB2 domains. All clusters revealed strong polarity preferences except for the DS clusters.

#### 2.3.3. Full PCPE-1

GAG binding was characterized with full-length PCPE-1 models obtained using UNRES CG simulations and HP dp6 as a ligand. Binding to Model 1, which was the most probable model among the ones obtained without the SAXS-based restraints, was significantly stronger than to Models 2, 3 and the SAXS Model (Table 5, Appendix A), as well as to the NTR domain (t-test, *p*-value < 0.05). Binding to Model 3 was also significantly stronger than to Model 2 and to the NTR domain. All clusters of HP dp6 solutions obtained for Model 1 were located in the region formed by the same residues of the NTR domain, the linker and the CUB1-CUB2 domains. For Model 2, the first cluster was located differently from the second and the third clusters. All the clusters correspond to the residues belonging predominantly to the NTR domain and the linker, but also partially to the CUB1-CUB2 domains. For Model 3, only NTR residues contributed to the binding of HP dp6. Cluster 1 was the most representative for Model 3 according to the molecular docking results, although not the most favorable according to MM-GBSA calculations, which again points to the essential differences in molecular docking and MD-based scoring. In the SAXS Model, GAG binding occurred at the NTR/linker interface. In all cases, the clusters were located in PCPE-1 patches corresponding to the positive electrostatic potential shown in Figure 2 and Appendix A. 

### 2.4. The Potential Role of Ca^2+^ in PCPE-1 Interactions with Glycosaminoglycans

#### 2.4.1. Prediction of Ca^2+^ Binding Sites

According to the experimental data, the interactions between both full-length PCPE-1 and the NTR domain with HP and HS are cation-dependent [28]. Therefore, we attempted to analyze the impact of Ca^2+^ ions on HP binding in silico, which allowed us to evaluate the available computational tools in terms of sensitivity and prediction power to account for divalent ions in such calculations. As a first step, we applied three different approaches (see Section 3.7 for details) to annexin V protein, which has 9 experimentally identified occupied Ca^2+^ binding sites, some of which are occupied upon HP binding [35]. The IonCom server predicted correctly eight out of nine experimentally known binding sites, while FoldX and MD approaches correctly predicted six binding sites (Table 6). 

Furthermore, we performed MM-GBSA calculations to estimate if the strength of the Ca^2+^ binding in these experimentally known binding sites correlated with the predictions (Table 7). As shown in the table, the total energies of interactions were positive despite the fact that all the ions were stable during the entire MD simulation performed in explicit solvent. This reflects the fact that the implicit continuous solvent model in MM-GBSA fails to properly account for the strength of binding for these divalent ions in terms of the full binding free energy. At the same time, in vacuo electrostatic energy was highly negative and could be meaningful for comparing binding sites since the studied interactions were electrostatically driven. A t-test performed for the in vacuo electrostatic energy values did not point out any statistical differences between the sites, which were properly predicted and the ones which the MD-based approach failed to predict.

We applied three ion-binding site prediction methods to the NTR and the CUB1-CUB2 domains of PCPE-1 (Table 6). Neither FoldX nor IonCom found any Ca^2+^ binding site for the NTR domain, while the MD approach identified from one to three binding sites, one of which being consistent through all five repetitions of MD simulations. The fact that these methods did not agree with MD simulations could be due to conformational changes of negatively charged amino acid side chains during the MD simulation, allowing them to come close to each other and to coordinate calcium ions. FoldX and IonCom used static structures, which prevents the dynamics required for the coordination of Ca^2+^. For CUB1-CUB2 domains, all methods were consistent and predicted two Ca^2+^ binding sites identical to those found in the CUB1-CUB2 domain complexed with procollagen (PDB ID: 6FZV). This means that CUB1-CUB2 domains in PCPE-1 could be already prebound to Ca^2+^ ions when the interaction with the procollagen is established. Furthermore, we compared the predicted Ca^2+^ binding sites for PCPE-1 domains in terms of electrostatic energies obtained from MM-GBSA calculations with the corresponding energies for annexin V in order to estimate their strength (Table 8). CUB1-CUB2 binding sites were energetically comparable with those of annexin V. The Ca^2+^ binding site in CUB2 was stronger than in CUB1 as suggested by more favorable electrostatic energies and by the fact that the Ca^2+^ binding site in CUB2 was identified by MD in all five MD replicas, while the Ca^2+^ binding site of CUB1 was correctly identified in three MD simulations. The Ca^2+^ binding sites predicted for the NTR domain were significantly weaker, and only one of them was found in all MD replicas. This suggests that this site may be unoccupied when the NTR domain is in solution and not bound to a GAG. The experimental evidence that the NTR domain binding to HP is dependent on divalent cations [28] leads to the hypothesis that Ca^2+^ ions could potentially bind within the interface of the NTR-HP complex.

We calculated electrostatic potential isosurfaces for the NTR and CUB1-CUB2 domains in the presence and in the absence of Ca^2+^ ions using the PBSA approach to predict how the electrostatic properties of the protein were affected by Ca^2+^ ions binding, which, in turn, could have an impact on GAG binding. For this, we used two Ca^2+^ binding sites corresponding to the X-ray structure (PDB ID: 6FZV) of the CUB1-CUB2 domain, and the weak Ca^2+^ binding site predicted in the NTR domain (Figure 5). Major differences in both positive and negative electrostatic potential shape were observed for the NTR domain. This is not only explained by the direct effect of Ca^2+^ positive charge but also by the fact that E405, E406 and N407 were moved closer to each other to coordinate the cation, which also affects the topology of the isosurface. For CUB1-CUB2 domains, the largest positively charged patch of the potential isosurface was not noticeably affected by the presence Ca^2+^ ions. To sum up, the predicted 3 Ca^2+^ binding sites for both PCPE-1 domains, when occupied, could potentially affect GAG binding. This potential effect is analyzed below. 

#### 2.4.2. PCPE-1 Interactions with Glycosaminoglycans in the Presence of Ca^2+^ Ions

In order to analyze the potential impact of Ca^2+^ ions on PCPE-1-GAG interactions, HP dp2, dp4 and dp6 were docked onto NTR, CUB1-CUB2 and the full-length protein in the presence of Ca^2+^ ions, followed by MD-based analysis. Two Ca^2+^ ions were prebound to CUB1-CUB2 and one to the NTR domain. Despite the differences in electrostatic properties of these domains described above, no significant changes in docking results or binding free energies were observed when compared with the domains without prebound Ca^2+^ (Appendix A and Appendix A). In both cases, the binding occurred in the regions distant from the Ca^2+^ ion binding sites. Only one cluster (HP dp6, cluster 3 in the NTR domain) was found to be close to the Ca^2+^ ion. In the corresponding binding poses the ion was coordinated by a sulfate group from the terminal GlcNS(6S) residue. However, the binding poses from this cluster were significantly less favorable than those located distantly from the Ca^2+^ ion.

We also analyzed the impact of Ca^2+^ ions on the GAG binding to the full-length PCPE-1 (Table 9, Appendix A, Figure 6, Appendix A). For Models 1 and 2 and the SAXS Model, we did not observe any effect of Ca^2+^ ions on the location of the structural clusters nor the direct participation of the Ca^2+^ ions in binding HP dp6. For Model 3, there was a significant difference between the clusters observed in the absence and in the presence of Ca^2+^ ions (e.g., cluster 2) related to the essential changes of the electrostatic potential on the protein surface in the presence of Ca^2+^. In terms of the free energies of binding, the presence of Ca^2+^ ions did not significantly affect binding to Models 1 or 2 or the SAXS Model. For cluster 2 of Model 3, which was relocated in the presence of Ca^2+^ ions, the free energy of binding became less favorable than in the absence of Ca^2+^ ions.

To summarize our attempts to determine the impact of Ca^2+^ on GAG binding to PCPE-1, our approach to consider Ca^2+^ ions as a part of the protein did not detect any favorable effect of these divalent ions on GAG binding in contrast to the experimental data [28]. Therefore, we hypothesize that Ca^2+^ ions might bind to GAGs rather than to PCPE-1 prior to the complex formation. The binding of divalent ions to GAGs has been experimentally reported for Zn^2+^, Mn^2+^, Cu^2+^, Ca^2+^, Co^2+^, Na^+^, Mg^2+^, Fe^3+^, Ni^2+^, Al^3+^ and Sr^2+^ ions [36,37]. The crucial role of cations in protein-GAG interactions was experimentally shown for amyloid precursor protein [38], HP cofactor II [39], endostatin [40,41], FGF1 and IL-7 [42]. Experimental data obtained using mass spectrometry [43], NMR [44], gel-filtration chromatography [45,46] and infrared spectroscopy [47] indicate that GAGs interact with divalent ions, and that these interactions affect GAG structure and conformational properties. Divalent ions will be integrated in GAG structure in our future work on protein-GAG complexes. Another potential role of Ca^2+^ ions could be to stabilize PCPE-1 structure, which would affect its interactions with GAGs. 

#### 2.4.3. Predicting Longer GAG Binding Poses Using the Fragment-Based Approach

We calculated the binding poses of long (dp11) HP chains on full-length PCPE-1 in the absence and the presence of three Ca^2+^ ions, two bound to CUB1-CUB2 and one to the NTR domain (Table 10, Appendix A). HP dp11 was the longest GAG that we managed to assemble in these docking experiments, and it was used to model the scenario when the GAGs longer than dp6 are bound to the protein. The increase in length of the HP chain from dp6 to dp11 significantly stabilized (*p*-value < 0.05) the interactions with PCPE-1 Models 1 and 2. The addition of Ca^2+^ strengthened the interactions of HP dp11 with PCPE-1 for Models 1, 2 and the SAXS Model but not for Model 3. Similarly to what was observed for HP dp6, Model 1 was the strongest in terms of HP dp11 binding. Ca^2+^ ions did not affect the binding sites of HP dp6 on the surface of Models 1, 2 or the SAXS Model but changed binding to Model 3, as described for HP dp6 (Figure 7, Appendix A). In all the cases, docked HP dp6 structures overlapped very well with those of HP dp11. It could be concluded that docking HP dp6 defines the core binding unit of a HP chain. The increase in binding affinity with the increase of HP length agrees with the experimental trend observed in this study (Figure 1).

## 3. Materials and Methods

### 3.1. Surface Plasmon Resonance (SPR) Binding Assays

The SPR measurements were performed on a BIAcore 3000 instruments (GE Healthcare, Uppsala, Sweden), and the data were analtzed with the BIAevaluation 3.1 Software (GE Healthcare, Uppsala, Sweden) as previously described [28]. Inhibition assays of PCPE-1 binding to HP and HS by HP oligosaccharides (from dp2 up to dp8, generous gift of Rabia Sadir and Hugues Lortat-Jacob, Institut de Biologie Structurale, Grenoble, France) were carried out as previously described [28]. Briefly, HP (Sigma, St Quentin Fallavier, France) and HS (Celsus Laboratories Inc, Cincinnati, OH, USA) from porcine intestinal mucosa were biotinylated and captured on streptavidin previously immobilized on a CM4 sensor chip (GE Healthcare, Uppsala, Sweden). Human recombinant PCPE-1 (1 µM) [28] was incubated with HP oligosaccharides (5 µg/mL) in 10 mM Hepes pH 7.5 + NaCl 0.15 M + P20 0.005% (HBS) + 5 mM CaCl_2_ for one hour before injection over immobilized HP (39 RUs) and HS (113 RUs) at a flow rate of 30 µL/min for 4 min. The percentages of inhibition were calculated relative to PCPE-1 binding level incubated in the same conditions with HBS + 5 mM CaCl_2_. The running buffer was HBS and the temperature was set at 25 °C.

### 3.2. Structures

#### 3.2.1. Protein Structures

The structure of the N-terminal CUB1-CUB2 domains was obtained from the X-ray structure of CUB1-CUB2 fragment of PCPE-1 bound to the C-propeptide trimer of procollagen III (PDB ID: 6FVZ, 2.7 Å). In this structure, the residues 33–275 are resolved (here and further, the numeration of the sequence corresponds to the UniProtKB ID: Q15113). However, the structure of the linker (151–157) between two CUB domains was not determined due to its flexibility [22]. The structure of the NTR domain (313–442) was also obtained from the PDB (PDB ID: 1UAP, 1st NMR model) [23]. The structures of the linker between two CUB domains (151–157) as well as the interdomain linker (276–312) were built in xLeap module of AMBER16, refined using an MD approach and then used for modeling the full-length structure of PCPE-1 with a CG MD approach as described in the Section 3.4.

#### 3.2.2. Glycosaminoglycan Structures

The following GAG structures were used for molecular docking: CS6 dp4, CS6 dp6, DS dp6, HP dp2, dp4, dp3, dp6 with (IdoA(2S) ring in ^1^C_4_ conformation, as this conformation is clearly predominant in HP [9,48]). These structures were built in our previous work [49]. 

### 3.3. Electrostatic Potential Calculations

The PBSA approach as implemented in AmberTools within AMBER16 package [50] was used to calculate electrostatic potential isosurfaces corresponding to the analyzed proteins. This method proved to be successful for GAG binding site prediction on an extensive protein-GAG dataset from the PDB [31]. The default value for the grid spacing of 1.0 Å and ff99SB force field parameters were used. The electrostatic potential isosurfaces were analyzed in VMD [51]. For visualization, such values of positive and negative electrostatic potential were chosen for each molecular system so that the data were as informative as possible for GAG binding site propensities. 

### 3.4. Coarse-Grained MD Simulations

In order to calculate the conformations of the PCPE-1 linkers (151–157 and 276–312 sequences) to obtain a model of the full-length protein, we applied a CG multiplexed replica exchange molecular dynamics (MREMD) [52,53] approach as implemented in UNRES (United Residues) [54,55], as previously described. The protocol was similar to that used in our previous work [12]. Distance restraints were imposed on the domains during the MREMD simulations. Additionally, in one of the simulations, SAXS-derived restraints [22] were imposed [56]. Each MREMD simulation consisted of 20 trajectories run at temperatures from 265 K to 370 K. Each trajectory consisted of 3.5 **×** 10^7^ MD steps with 4.89 fs length for simulations with the restraints on the domains only and 7.1 **×** 10^4^ steps for simulations with additional SAXS-derived restraints. The lower number of steps for simulations with information from the SAXS experiment was used because the radius of gyration of maintained structures was obtained already after that time. Only conformations from the last quarter of the simulation were taken into further analysis with the use of the weighted histogram analysis method (WHAM) [56]. The next step was minimum variance cluster analysis [57] of the conformational ensemble at T = 300 K, which enabled us to obtain five clusters, ranked according to summary probabilities of the ensembles and containing the most probable structures to the cluster with the least probable structures. For each cluster, one representative structure, closest to the cluster centroid, was selected as the representative conformation. The last step was the conversion of CG structures into all-atom ones using the PULCHRA [58] and SCWRL [59] algorithms.

### 3.5. Molecular Docking

#### 3.5.1. Autodock 3

The docking simulations of GAG ligands to the PCPE-1 were performed with Autodock 3 (AD3) [60], which was previously shown to yield the best performance among docking programs for GAG ligands [10,31]. The blind docking procedure was used: the whole protein surface was available for the ligand when sampling a potential binding site. For all proteins, we used 127 **×** 127 **×** 127 grid points for AD3 runs. However, because of the differences in protein size, the grid step was different for different proteins to contain the whole protein molecule within a single grid box: the default value of 0.375 Å was used for the NTR domain, while 0.5 Å was used for CUB1-CUB2 domains, and the grid step of 0.6–1.0 Å was used for different models of the full-length PCPE-1 protein due to their essentially bigger sizes. All GAG ligands were docked to the protein with the use of the Lamarckian genetic algorithm. The initial population size was 300, 10^5^ generations, 9995·10^5^ energy evaluations and 1000 independent runs with up to 33 torsional angle degrees of freedom were carried out. 1000 docked structures for each molecular system (protein-GAG pair) were obtained and further analyzed; the 50 top-scored ones (according to AD3 scoring function) were chosen for further clustering with the DBSCAN algorithm [33]. The clustering parameters, neighborhood search radii and minimal numbers of cluster members were manually selected for each system individually in order to yield 2‒4 representative clusters. The distance metric used for clustering, which is defined as the root-mean-square of atomic distances for the nearest atoms of the same type, takes into account the periodic nature of GAGs, which is more appropriate for those ligands than the classical root-mean-square deviation (RMSD) [61]. Within each cluster, those poses which were different from one another to be categorized into subgroups were selected for further analysis. Such a procedure was used to account for the multiple pose binding previously observed for GAG ligands [11,62]. The GAG glycosidic linkages from the obtained docking poses were visually filtered in order to avoid incorrect geometries that could be produced by AD3 [63]. 

#### 3.5.2. Fragment-Based Approach

In order to dock longer HP to the full-length PCPE-1 protein, which is unfeasible for the AD3 protocol described above due to the limitation of the available number of the degrees of freedom and, in general, because of the computationally very expensive conformational sampling required for simulations with such ligands, a fragment-based docking approach we recently developed was applied [64]. In brief, first, HP dp3 of both types, IdoA(2S)-GlcNS(6S)-IdoA(2S) and GlcNS(6S)-IdoA(2S)-GlcNS(6S), were docked using the AD3 docking procedure described in Section 3.5.1. Then, all 1000 solutions for each HP dp3 fragments were used to assemble ~10^5^ dp11 GAG chains applying the approach standard parameters [64]. This was followed by the refinement and all-atom conversion of the chains with a slight modification in the original scripts to avoid the RMSD-based selection of the best fitting structures compared to the experimental ones due to the lack of proper atomistic experimental PCPE-1/GAG complex structures. In particular, instead of the previously described way of selecting structures, a simple RMSD-based clustering (cutoff 4 Å) was performed to filter out the duplicates and find the most relevant structures. From the resulted ~5‒40 atomistic structures the ones with significantly different docking poses were selected and refined together with the full protein using MD simulations applying the same procedure as described in Section 3.6, except for the minimization procedure where at the first step of 10^4^–10^5^ steepest descent minimization cycles were applied before the conjugate gradient minimization step.

### 3.6. All-Atom MD Simulations and MM-GBSA Free Energy Calculations

All-atom molecular dynamics (MD) simulations of the PCPE-1, PCPE-1/Ca^2+^ and PCPE-1/Ca^2+^/GAG complexes obtained by molecular docking were performed with the use of the AMBER16 MD package [50]. Periodic boundary conditions in a truncated octahedron TIP3P water box with at least 8 Å distance from the solute to the periodic box border were used. Na^+^ and Cl^−^ monovalent counterions were used to neutralize the system. ff99SB force field parameters for protein [65] and the GLYCAM06 [66] for GAGs were used, respectively. Prior to MD production runs, two energy-minimization steps were performed: first, 500 steepest descent cycles and 1000 conjugate-gradient cycles with harmonic force restraints on solute (10 kcal/mol/Å^2^), then, 3000 steepest-descent cycles and 3000 conjugate-gradient cycles without restraints. After the minimization, the system was heated up to 300 K for 10 ps with harmonic force restraints on solute (10 kcal/mol/Å^2^), equilibrated for 100 ps at 300 K and 10^5^ Pa in isothermal isobaric ensemble (NTP). This was followed by a 100 ns MD production run in the same NTP ensemble. The SHAKE algorithm, 2 fs time integration, 8 Å cutoff for non-bonded interactions, and the particle mesh Ewald method were used. The trajectories were analyzed using the cpptraj module of AMBER Tools [46]. Free-energy calculations and per-residue energy decomposition were done using molecular mechanics-generalized born surface area (MM-GBSA) model igb = 2 [67] for protein-GAG and protein-Ca^2+^ complexes for the parts of the trajectory where convergence in terms of RMSD was obtained. The obtained energy values account explicitly for the enthalpy and implicitly for the solvent entropy. For this reason, the reported energies should not be strictly interpreted as full free energy of binding: the entropic contribution to binding was not taken into account explicitly. Entropy calculations were shown to dramatically increase the overall noise in the free binding energies when used within MM-GBSA free energy calculation schemes in general [68] and for GAG containing systems particularly [69].

### 3.7. Ca^2+^ Ion Position Prediction

We applied and compared several approaches to predict the Ca^2+^ binding sites on the surface of PCPE-1 and its domains. Prior to applying these approaches to PCPE-1, we analyzed their performance for the complex between an annexin V and HP, where Ca^2+^ ions are known to be stable and to contribute to GAG binding (PDB ID: 1G5N, 2.7 Å) [35]. In this structure, 9 Ca^2+^ were resolved. 

#### 3.7.1. Molecular Dynamics Approach

We used the MD approach with the protocols described in Section 3.6 to predict the binding sites of Ca^2+^ on the surface of protein. The length of each MD simulation run was 100 ns. In these calculations, Ca^2+^ ions were placed randomly in the periodic box. For annexin V, 9 Ca^2+^ were used corresponding to the number of Ca^2+^ ions observed in the experimental structure. For CUB1-CUB2, NTR domains, three ions were used. The number of ions was chosen in order to sample effectively the protein surface within a reasonable simulation time. For the CUB1-CUB2 and the NTR domains, the simulations were repeated 5 times. The trajectories were analyzed, and the for the frames where the Ca^2+^ ions were stably bound in terms of RMSD convergence for the coordination complex, MM-GBSA free energy calculations were performed. The obtained values for the predicted Ca^2+^ binding sites were compared with the corresponding energies obtained from the simulations of annexin V-Ca^2+^ and CUB1-CUB2-Ca^2+^ crystal structures. The latter was extracted from the structure of the complex of CUB1-CUB2 with procollagen peptide (PDB ID: 6FVZ, 2.7 Å). 

#### 3.7.2. FoldX and IonCom

We used the scripts of FoldX available at http://foldx.embl.de and online ion ligand binding site prediction tool IonCom at https://zhanglab.ccmb.med.umich.edu/IonCom for Ca^2+^ binding site predictions. FoldX represents a tool with an implemented empirical force field developed for effective evaluation of the contribution of mutations on the stability, folding and dynamics of proteins and nucleic acids [70,71]. As an output FoldX yields the positions of predicted ions on the surface of the protein. In contrast, IonCom utilizes ab initio training and template-based information to output a list of protein residues potentially involved in ion binding [72]. Both programs were used with the default parameters.

### 3.8. Visualization and Data Analysis

VMD [51], Chimera [73] and Pymol [74] were used for structural analysis visualization, MD trajectory analysis, as well as for the graphics production. R package was used for data analysis [75].

## 4. Conclusions

We report here the computational analysis of the interactions of full-length PCPE-1 and its domains with GAGs of various lengths and sequences. This model of full-length PCPE-1 based on SAXS restraints is in agreement with the experimental values of its radius of gyration and length. The full-length protein binds GAGs through the NTR domain and the interdomain linker, while the binding to CUB1-CUB2 domains is weaker, likely non-specific, and less energetically favorable. GAG preferential binding to the NTR domain is mostly electrostatically-driven. CS6 is predicted to bind to a different site of the NTR domain than the other GAGs, which may account for the experimental differences previously observed between CS6 and DS/HP [28]. Fragment-based docking of longer GAG oligosaccharides results in overlap with the docking poses obtained for shorter GAGs and corresponds to more favorable interactions than those established by shorter oligosaccharides in agreement with the strongest inhibition of PCPE-1-HP interactions by longer HP oligosaccharides reported in this study. Our results suggest that calcium ions may bind to GAGs before they interact with PCPE-1 or may stabilize the structure and conformation of full-length PCPE-1. Although we have used several computational approaches to predict Ca^2+^ binding sites on the protein surface, considering calcium ions as a part of the protein receptor for docking is not an approach applicable to all systems.

From a methodological point of view, we have shown that the size of the clusters identified by molecular docking is not correlated with their free binding energies obtained in the MD simulation. We have successfully applied, for the first time, fragment-based docking to dp11 oligosaccharides, which will be useful for the computational characterization of protein interactions with long GAGs, which are challenging to study using conventional docking approaches.

## Figures and Tables

**Figure 1 ijms-20-05021-f001:**
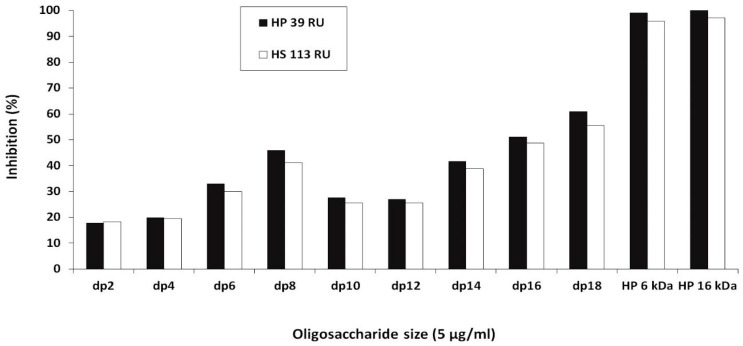
Surface plasmon resonance (SPR) inhibition assays. Inhibition of the binding of recombinant human procollagen C-proteinase enhancer-1 (PCPE-1) to biotinylated heparin (HP) and heparan sulfate (HS) captured on a streptavidin sensor chip (39 and 113 resonance units (RU) respectively) by HP oligosaccharides of different degrees of polymerization (dp2‒dp18) and by HP (6 and 16 kDa) at a concentration of 5 µg/mL.

**Figure 2 ijms-20-05021-f002:**
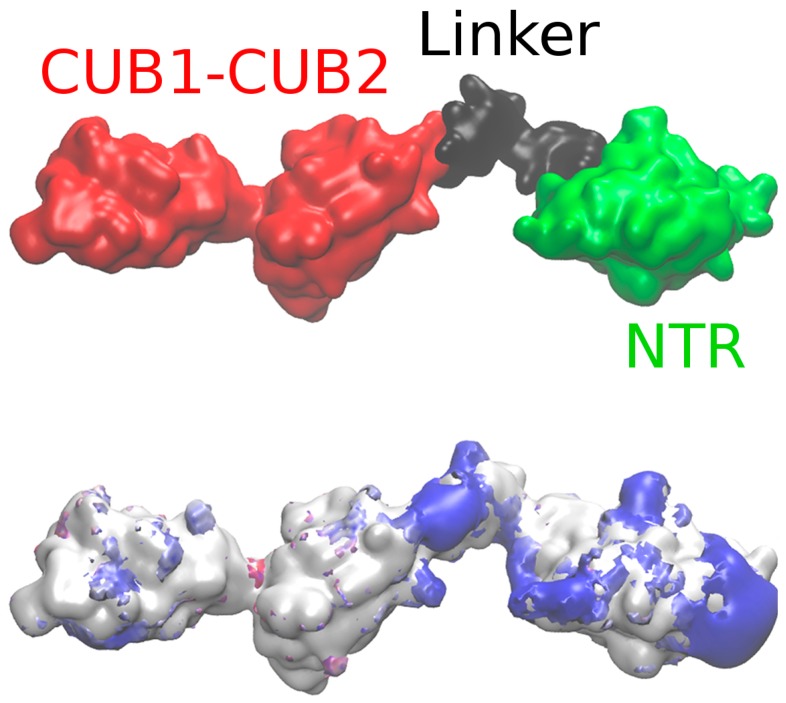
Small angle X-ray scattering (SAXS) Model (upper panel). Netrin-like (NTR) domain: green; CUB1-CUB2: red; the interdomain linker between the CUB2 and NTR domains: black. Positive electrostatic potential isosurfaces (2.0 kcal/mol · e^−1^) in the absence of Ca^2+^ ions obtained by Poisson‒Boltzmann surface area (PBSA) calculations (bottom panel).

**Figure 3 ijms-20-05021-f003:**
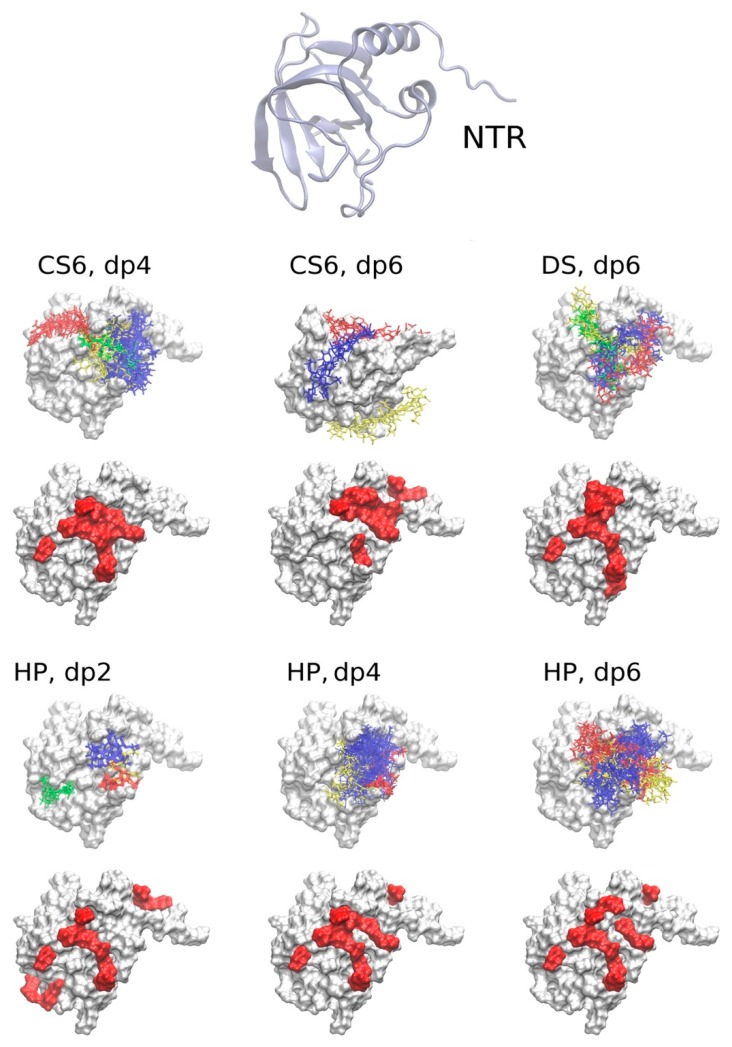
Molecular docking and molecular mechanics-generalized born surface area (MM-GBSA) for NTR-glycosaminoglycan (GAG) complexes. The structure of the NTR domain is shown in cartoon representation at the top. For each GAG, the analyzed clusters of docking solutions are shown in blue, red, yellow and green (from the most to the less populated cluster); the top 10 residues binding to GAGs according to MM-GBSA calculations averaged per GAG are highlighted in red surface. Note that the clusters for CS6 dp6 are shown for a different protein spatial orientation to allow for a better visualization. In addition, averaging the per-residue energy for very different clusters could be misleading as shown for CS6 dp6: the residues shown in red do not overlap with the surface patches where the most representative clusters of solutions are located.

**Figure 4 ijms-20-05021-f004:**
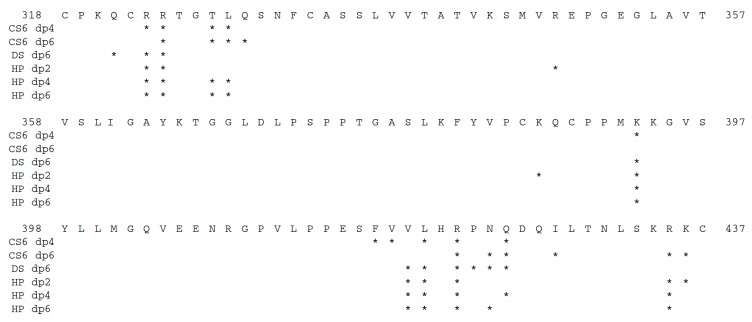
NTR amino acid residues identified in the top 10 for binding GAGs according to MM-GBSA calculations per cluster are labeled as an asterisk.

**Figure 5 ijms-20-05021-f005:**
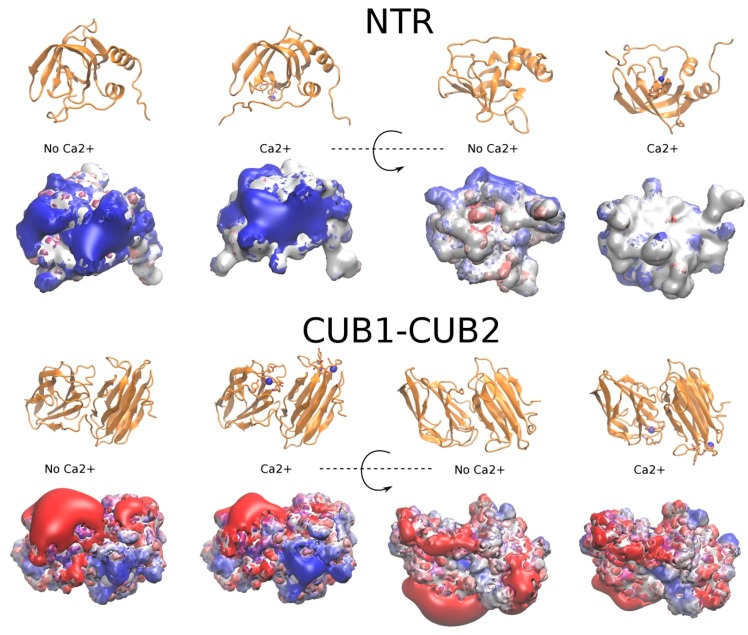
Electrostatic potential isosurfaces (blue, positive; red, negative) of NTR (–2.5 kcal/mol·e^−1^ and 1.0 kcal/mol·e^−1^) and CUB1-CUB2 (–3 kcal/mol·e^−1^ and 3 kcal/mol·e^−1^) domains in the presence and in the absence of Ca^2+^ ions obtained by PBSA calculations. Protein domains are shown in cartoon with the residues coordinating Ca^2+^ ions in licorice representation; Ca^2+^ ions: blue spheres.

**Figure 6 ijms-20-05021-f006:**
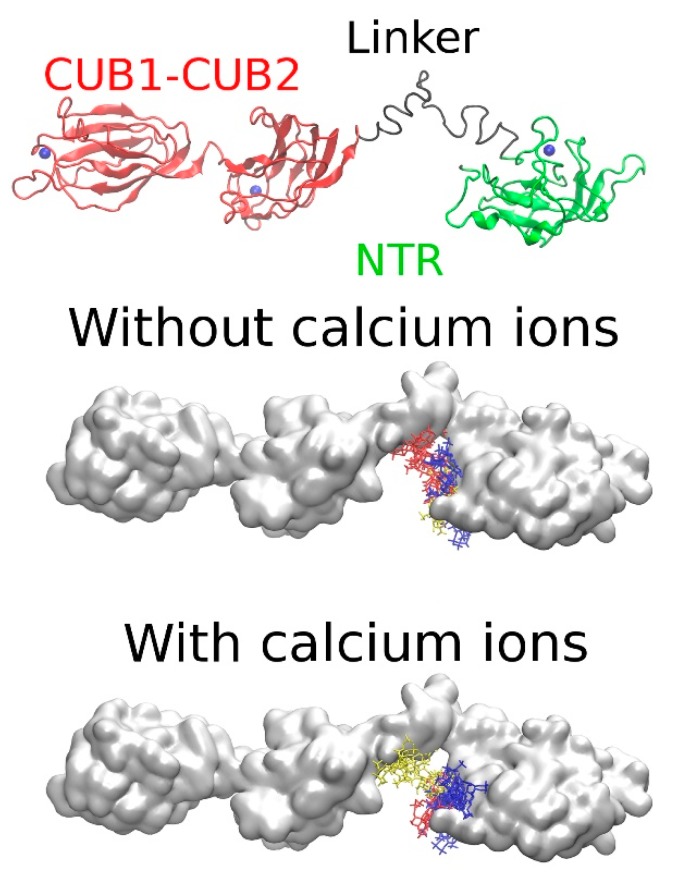
Molecular docking results for the models of the full-length PCPE-1 SAXS Model in the absence and presence of Ca^2+^ ions and HP dp6. The clusters of docking solutions are shown in blue, red and yellow (from the most to the least populated clusters). NTR domain: green; CUB1-CUB2: red; the interdomain linker between the CUB2 and NTR domains: black.

**Figure 7 ijms-20-05021-f007:**
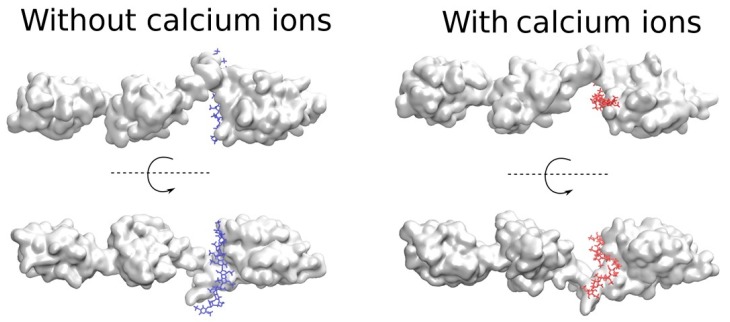
Molecular docking results for the models of the full PCPE-1 SAXS Model in the absence (in blue) and the presence (in red) of Ca^2+^ ions and HP dp11 corresponding to the most favorable free binding energies.

**Table 1 ijms-20-05021-t001:** Models of the full-length PCPE-1 obtained using UNRES coarse-grained (CG) simulations.

Model	Restraints	Probability	Radius of Gyration (Å)
1	CUB1-CUB2, NTR domains	34	22.2
2		32	24.8
3		18	22.6
4		8	22.8
5		8	22.6
1	CUB1-CUB2, NTR domains + SAXS-based	39	43.5
2		21	44.5
3		17	43.5
4		14	43.3
5		9	43.9

**Table 2 ijms-20-05021-t002:** Molecular docking molecular dynamics (MD)-based analysis summary for NTR-GAG interaction.

GAG	^1^ m, ε	^2^ #	^3^ Size	^4^ ΔG (kcal/mol)	^5^ Polarity
CS6, dp4	3, 2	1	19	−42.0 ± 6.6; −48.3 ± 7.7; −41.3 ± 6.6	17/2
		2	6	−30.1 ± 16.0; −63.3 ± 7.1	6/0
		3	4	−34.4 ± 9.6; −38.4 ± 8.6	2/2
		4	3	−46.7 ± 10.5	3/0
CS6, dp6	3, 2	1	3	−56.6 ± 9.0	3/0
		2	3	−33.9 ± 9.2	3/0
		3	3	−36.8 ± 7.1; −64.2 ± 11.8	3/0
DS, dp6	3, 2	1	6	−35.5 ± 6.3; −41.5 ± 6.8	5/1
		2	4	−36.7 ± 6.6	4/0
		3	3	−63.7 ± 8.3	3/0
		4	3	−37.8 ± 8.2	2/1
HP, dp2	3, 2	1	25	−44.9 ± 9.3; −41.1 ± 7.3; −23.1 ± 7.6	25/0
		2	12	−27.9 ± 9.2	12/0
		3	9	−42.0 ± 9.0	9/0
		4	3	−27.7 ± 8.9; −28.7 ± 5.6	3/0
HP, dp4	3, 2	1	32	−39.0 ± 7.2; −29.4 ± 10.4	21/11
		2	3	−53.9 ± 7.2	3/0
		3	3	−50.6 ± 11.5; −57.4 ± 8.6	2/1
HP, dp6	3, 2	1	15	−69.5 ± 7.8; −56.7 ± 7.4; −43.5 ± 9.7; −54.0 ± 14.8; −80.5 ± 10.7	9/6
		2	7	−68.3 ± 11.0; −44.7 ± 7.5; −57.1 ± 10.6; −55.4 ± 9.1	4/3
		3	6	−50.1 ± 10.0; −44.6 ± 9.5; −65.7 ± 11.5; −61.8 ± 13.6	4/2

^1^ DBSCAN parameters *m*, the minimal neighborhood size, and ε, neighborhood search radius [33]; ^2^ cluster number; ^3^ cluster size (number of solutions); ^4^ free energy of binding obtained by MM-GBSA; ^5^ the polarity of a GAG binding pose was defined as its preferred orientation in relation to the reducing and non-reducing end.

**Table 3 ijms-20-05021-t003:** Similarity of GAG binding poses for the NTR domain as of common amino acid residues identified in the top 10 for binding according to MM-GBSA calculations per cluster.

GAG	CS6, dp4				CS6, dp6			DS, dp6				HP, dp2				HP, dp4			HP, dp6		
CS6, dp4	10	7	7	6	6	4	7	5	6	6	4	7	5	5	5	6	6	6	7	7	7
	7	10	6	9	3	2	6	6	6	7	5	6	7	6	6	5	8	7	8	7	9
	7	6	10	5	4	3	5	6	5	7	4	7	4	6	5	4	5	7	7	7	6
	6	9	5	10	3	2	5	6	6	6	4	5	7	5	5	6	9	6	7	6	8
CS6, dp6	6	3	4	3	10	7	5	2	5	3	1	4	3	1	1	5	3	2	5	5	4
	4	2	3	2	7	10	5	2	4	2	1	5	2	1	1	4	2	1	4	3	3
	7	6	5	5	5	5	10	5	5	4	5	6	5	4	4	7	5	4	7	6	7
DS, dp6	5	6	6	6	2	2	5	10	5	6	6	6	6	7	6	5	6	8	7	7	7
	6	6	5	6	5	4	5	5	10	5	5	4	6	4	4	6	7	5	7	6	6
	6	7	7	6	3	2	4	6	5	10	6	6	6	6	6	3	5	7	7	8	6
	4	5	4	4	1	1	5	6	5	6	10	4	4	5	5	3	4	6	5	5	5
HP, dp2	7	6	7	5	4	5	6	6	4	6	4	10	5	5	5	5	5	6	7	7	7
	5	7	4	7	3	2	5	6	6	6	4	5	10	5	5	7	7	5	7	7	7
	5	6	6	5	1	1	4	7	4	6	5	5	5	10	6	3	5	7	6	6	6
	5	6	5	5	1	1	4	6	4	6	5	5	5	6	10	3	5	6	6	6	6
HP, dp4	6	5	4	6	5	4	7	5	6	3	3	5	7	3	3	10	7	3	6	5	6
	6	8	5	9	3	2	5	6	7	5	4	5	7	5	5	7	10	6	7	6	8
	6	7	7	6	2	1	4	8	5	7	6	6	5	7	6	3	6	10	6	7	7
HP, dp6	7	8	7	7	5	4	7	7	7	7	5	7	7	6	6	6	7	6	10	9	9
	7	7	7	6	5	3	6	7	6	8	5	7	7	6	6	5	6	7	9	10	8
	7	9	6	8	4	3	7	7	6	6	5	7	7	6	6	6	8	7	9	8	10

Each line/column in front/below each GAG reflects a separate cluster, for which average values were taken into account.

**Table 4 ijms-20-05021-t004:** Similarity of GAG binding poses for the NTR domain as of the number of common amino acid residues identified in the top 10 for binding according to MM-GBSA calculations per GAG.

GAG	CS6, dp4	CS6, dp6	DS, dp6	HP, dp2	HP, dp4	HP, dp6
CS6, dp4	10	5	7	6	9	8
CS6, dp6	5	10	4	4	6	6
DS, dp6	7	4	10	6	7	7
HP, dp2	6	4	6	10	7	7
HP, dp4	9	6	7	7	10	9
HP, dp6	8	6	7	7	9	10

**Table 5 ijms-20-05021-t005:** Molecular docking MD-based analysis summary for PCPE-1 SAXS Model/HP dp6 interaction.

^1^ m, ε	^2^ #	^3^ Size	^4^ ΔG, kcal/mol	^5^ Top_MM-GBSA_ 10 Residues for GAG Binding	^6^ Polarity
2, 2.64	1	4	−62.4.8 ± 19.0; −54.9 ± 9.1 −49.6 ± 18.6	R435, K436, R275, R288, K279, K299, K365, K434, N331, K295	4/0
	2	3	−50.1 ± 9.7; −79.0 ± 17.0; −38.1 ± 9.4	K436, R435, K365, K299, K434, K271, K295, R288, K165, K279	3/0
	3	3	−30.8 ± 10.7; −36.0 ± 7.8; −42.3 ± 10.6	K299, K436, K279, K365, K271, K434, K295, K165, Q282, R435	2/1

^1^ DBSCAN parameters *m*, the minimal neighborhood size, and ε, neighborhood search radius [33]; ^2^ cluster number; ^3^ cluster size; ^4^ free energy of binding obtained by MM-GBSA; ^5^ residues identified in the top 10 for binding according to MM-GBSA calculations per cluster ordered by the impact (starting from the most favorable one). ^6^ The polarity of a GAG binding pose was defined as its preferred orientation in relation to the reducing and non-reducing end.

**Table 6 ijms-20-05021-t006:** Ca^2+^ predictions for annexin V and PCPE-1 domains: number of the binding sites predicted are provided.

Protein	PDB ID	Experimental Structure	Method		
			FoldX	IonCom	^1^ MD
Annexin V	1G5N	9	6	8	6
NTR	1UAP	0	0	0	2
					3
					1
					1
					1
CUB1-CUB2	6FZV	2	2	2	2
					2
					1
					2
					1

^1^ Five repetitions of the MD simulations were performed for PCPE-1 domains.

**Table 7 ijms-20-05021-t007:** MM-GBSA free energy calculations (per Ca^2+^ ion) for the experimentally known Ca^2+^ binding sites in annexin V.

Ca^2+^ Number (X-Ray)	^1^ ΔG, kcal/mol	^2^ ΔG_ele_, kcal/mol	^3^ FoldX	^3^ IonCom	^3^ MD
319	57.2 ± 4.7	−310.4 ± 10.3	+	+	+
320	47.5 ± 4.8	−264.8 ± 15.7	+	+	–
321	36.5 ± 3.5	−296.0 ± 10.9	–	+	+
322	59.7 ± 4.9	−380.5 ± 9.5	+	+	–
323	36.4 ± 3.5	−332.4 ± 7.9	–	–	+
324	62.4 ± 4.4	−376.6 ± 8.1	+	+	+
325	47.5 ± 6.1	−413.2 ± 13.0	+	+	+
326	39.3 ± 3.7	−312.2 ± 9.2	–	+	+
327	59.2 ± 4.7	−302.3 ± 8.6	+	+	–

^1^ and ^2^: ΔG and ΔG_ele_ stand for the total and in vacuo electrostatic MM-GBSA free energies, respectively. ^3^ Plus and minus reflect whether the method was capable of predicting the corresponding experimentally detected binding site correctly.

**Table 8 ijms-20-05021-t008:** MM-GBSA free energy calculations (per Ca^2+^ ion) for the predicted Ca^2+^ binding sites in PCPE-1 domains and corresponding Ca^2+^ binding site occupancy in 100 ns MD simulation.

PCPE-1 Domain	Ca^2+^ Site	^2^ ΔG_ele_, kcal/mol	Site Occupancy, ns
NTR, MD1	E405, E406, N407 G367, D370	−116.5 ± 20.4 −58.3 ± 15.7	65 40
NTR, MD2	E405, E406, N407 D314/N-terminus of NTR G367, D370	−125.8 ± 14.7 −38.6 ± 19.0 −49.6 ± 14.2	85 35 90
NTR, MD3	E405, E406, N407	−120.3 ± 15.9	75
NTR, MD4	E405, E406, N407	−123.4 ± 12.2	45
NTR, MD5	E405, E406, N407 ^1^ E405, E406, N407/G367, D370	−51.7 ± 13.7; −156.2 ± 33.6	65 25
CUB1-CUB2 (X-ray, PDB ID: 6FZV)	E85, Y92, D93, D134 Y180, E208, D216, D258	−363.2 ± 10.8 −466.5 ± 12.5	100 100
CUB1-CUB2, MD1	E85, Y92, D93, D134 Y180, E208, D216, D258	−389.9 ± 19.3 −302.9 ± 24.3	25 90
CUB1-CUB2, MD2	E85, Y92, D93, D134 Y180, E208, D216, D258	−368.3 ± 18.7 −371.9 ± 11.8	85 90
CUB1-CUB2, MD3	Y180, E208, D216, D258	−389.2 ± 18.6	85
CUB1-CUB2, MD4	E85, Y92, D93, D134 Y180, E208, D216, D258	−293.7 ± 9.0 −374.0 ± 16.1	75 95
CUB1-CUB2, MD5	Y180, E208, D216, D258	−521.4 ± 10.8	95

^1^ In the course of this simulation, G367 and D370 moved towards E405, E406 and N407 to coordinate Ca^2+^. (MD: molecular dynamics, 1–5: replicas). ^2^ ΔG_ele_ stands for the in vacuo electrostatic MM-GBSA free energy.

**Table 9 ijms-20-05021-t009:** Molecular docking MD-based analysis summary for PCPE-1 SAXS Model/Ca^2+^/HP dp6 interaction.

^1^ m, ε	^2^ #	^3^ Size	^4^ ΔG, kcal/mol	^5^ Top_MM-GBSA_ 10 Residues for GAG Binding	^6^ Polarity
2, 2.8	1	6	−58.8 ± 12.2; −56.0 ± 19.7; −58.6 ± 13.7	R435, K436, K434, K365, K299, K279, K295, R288, P438, K271	5/1
	2	4	−79.5 ± 15.6; −32.6 ± 11.0; −42.8 ± 10.0	K436, R435, K279, R288, K365, K299, K434, Q282, G281, K287	3/1
	3	3	−30.8 ± 10.7; −70.7 ± 13.2; −56.6 ± 18.4	R435, K299, K436, K365, K434, R275, K279, K295, K271, K305	3/0

^1^ DBSCAN parameters *m*, the minimal neighborhood size. and ε, neighborhood search radius [33]; ^2^ cluster number; ^3^ cluster size; ^4^ free energy of binding obtained by MM-GBSA; ^5^ residues identified in the top 10 for binding according to MM-GBSA calculations per cluster ordered by the impact (starting from the most favorable one). ^6^ The polarity of a GAG binding pose was defined as its preferred orientation in relation to the reducing and non-reducing end.

**Table 10 ijms-20-05021-t010:** Fragment-based molecular docking MD analysis summary for PCPE-1 SAXS Model/Ca^2+^/HP dp11 interaction.

^1^ #	^2^ Ca^2+^	^3^ ΔG, kcal/mol	^4^ Top_MM-GBSA_ 10 Residues for GAG Binding
1	–	−65.6 ± 12.3	K299, R288, R435, K436, K295, K293, K305, K287, K365, P298
2	–	−64.2 ± 11.2	K436, K434, R275, K279, K295, R435, K365, R288, K299, K287
3	–	−94.7 ± 12.4	K295, K436, R435, K365, K434, K293, K299, R288, K305, V294
1	+	−93.3 ± 12.3	R435, K279, K295, K436, K305, K299, K434, N331, K271, R324
2	+	−73.8 ± 11.9	R435, K434, K436, K299, K295, K279, P298, K293, K287, K305
3	+	−102.5 ± 14.7	K436, K299, K295, R435, R275, K293, K279, K434, K305, P441

^1^ Pose number; ^2^ Ca^2+^ presence; ^3^ free energy of binding obtained by MM-GBSA; ^4^ residues identified in the top 10 for binding according to MM-GBSA calculations per cluster ordered by the impact (starting from the most favorable one).

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
