# Peer review of "Analysis of Procollagen C-Proteinase Enhancer-1/Glycosaminoglycan Binding Sites and of the Potential Role of Calcium Ions in the Interaction"

_ijms, 2019, doi:10.3390/ijms20205021_

Round 1

Reviewer 1 Report

This work discussed the interactions between PCPE-1 and GAGs. 

Too many keywords which should be limited to 5; In figure 1, the author said "There was an increase in inhibition with the length of HP 84oligosaccharides", however, it is decrease from dp8 to dp10 and dp 12. Could you explain this? From figure 1, we could see the results from dp2 to dp18, when you do the analyze work, only CS6 dp4, CS6 dp6, DA dp6, HP dp2, HP dp4 and HP dp6 were chosen as the modle. Why you didn't analyze series of GAGs?   For all the tables and figures, the format should be uniform; The conclusion part is too long, it must be rewrote.

Author Response

POINT 1. Too many keywords which should be limited to 5.

ANSWER. As requested by the reviewer we have limited the number of the keywords to 5: “Keywords: procollagen C-proteinase enhancer-1; glycosaminoglycans; computational analysis of protein-glycosaminoglycan interactions; calcium ions; fragment-based docking.

POINT 2. In Figure 1, the author said "There was an increase in inhibition with the length of HP oligosaccharides", however, it is decrease from dp8 to dp10 and dp 12. Could you explain this?

ANSWER. The explanation has been added in the revised manuscript as follows: “Here, we investigated the effect of HP oligosaccharides of various length as inhibitors of PCPE-1 binding to HP in order to determine the optimal size of HP required to bind to PCPE-1. There was a trend towards an increase in inhibition of PCPE-1-HP interaction with the length of HP oligosaccharides from dp2 to dp8, and then from dp14 to full-length HP chains (Figure 1). HP decasaccharides and dodecasaccharides (dp 10 and dp12, respectively) inhibited the binding of PCPE-1 to HP to a lesser extent than the HP octasaccharide (dp8). The oligosaccharides used for inhibition experiments were separated according to their size and not to their sulfation pattern and/or charges. They thus contain a mixture of oligosaccharides of the same size displaying different number of sulfate groups in different positions of their sequences resulting in different binding motifs with likely different inhibitory efficiencies. This heterogeneity might be more pronounced in dp10 and dp12 leading to a lower global inhibition by these oligosaccharides than by the octasaccharide.

POINT 3. From Figure 1, we could see the results from dp2 to dp18, when you do the analyze work, only CS6 dp4, CS6 dp6, DS dp6, HP dp2, HP dp4 and HP dp6 were chosen as the model. Why you didn't analyze series of GAGs?

ANSWER. The reasons for the choice of the GAGs analyzed in this study were the following: 1. we aimed to use the GAGs studied in the previous work (reference 28) in order to be able to make a direct comparison with those experimental data and to systematically evaluate the effects of epimerization, length and sulfation pattern of GAGs on binding; 2. conventional docking approaches are severely limited in terms of GAG size and can be used for length up to dp6 (reference 31). Therefore, to study the impact of the GAG length we used the series of HP dp2, dp4, dp6, while HP is the strongest binder and, therefore, the results for HP of different lengths are expected to be the most representative. The fragment-based method (reference 65), a recent novel approach developed for HP, can be used to dock longer HP molecules. Therefore, we applied it, for the first time, to the longest HP chains that were feasible to assemble for this protein target (dp 11).

We explain this choice of GAGs in more details page 4 of the revised manuscript:

These GAGs were selected for the following reasons: i) to compare the in silico data with the experimental ones previously obtained with these GAGs [24, and to investigate the effects of epimerization, length and sulfation pattern of GAGs on binding, and ii) conventional docking approaches are severely limited in terms of the size of GAGs and can be effectively used only for the GAGs length up to dp6 [31]. Therefore, we used HP oligosaccharides of different length, from dp2 to dp6, to determine the effect of the GAG length on the binding to PCPE-1. HP being the strongest binder, the results obtained with HP oligosaccharides of different length should be the most representative. Furthermore, the GAGs studied here were selected in order to systematically evaluate the changes in binding to PCPE-1 according to the GAG length (dp4-dp6 for CS6 and dp2-dp6 for HP), the epimerization of glucuronic acid (CS6 dp6 and DS dp6), the increase in the number and position of sulfated groups (i.e. the sulfation pattern) and the net charge of the oligosaccharides (CS6, DS and HP).

POINT 4. For all the tables and figures, the format should be uniform.

ANSWER. We have corrected this by providing the uniform formatting throughout the whole revised manuscript.

POINT 5. The conclusion part is too long, it must be rewrote.

ANSWER. We have substantially shortened the conclusion as requested by the reviewer.

Reviewer 2 Report

Jan Potthoff and coworkers describe in the manuscript "Analysis of procollagen C-proteinase enhancer 1/glycosaminoglycan interactions and a potential role of calcium ions in their mediation" a molecular modeling study intending to build a structural model of full-length procollagen C-proteinase enhancer-1 (PCPE-1) and then to predict binding sites of glycosaminoglycans (GAGs).

The molecules studied in this manuscript play important biological roles and, in the same time, display a lack of precise structure definition in particular because of the intrinsic flexibility of oligosaccharides on one side, and of the linkers connecting various domains in PCPE-1 on the other side. The molecular modeling study of Potthoff and coworkers is thus an essential step to be able to study the interactions between PCPE-1 and GAGs.

The model of PCPE-1 is built using several modeling techniques, as coarse-grained approach to investigate the linker structure, or the docking of GAG ligands using Autodock. The interaction energies are calculated using the MMPBSA approach. The positions of Calcium ions around PCPE-1 are compared in the presence and in the absence of Calcium ions.

The manuscript contains many results about the modeling of PCPE-1 and of their interactions with GAGs and Calcium ions. The modeling results are validated by comparison with SAXS measurements and measurements of inibition of PCPE-1 by oligosaccharides.

The work looks interesting and well conducted. Nevertheless, the quantity of validating  experimental information is small with respect to the quantity of modeling results. It would be interesting to better connect these two types of data in the manuscript.

Author Response

POINT 1. The work looks interesting and well conducted. Nevertheless, the quantity of validating experimental information is small with respect to the quantity of modeling results. It would be interesting to better connect these two types of data in the manuscript.

ANSWER. To elaborate this point, we have stressed the comparison with the experimental data from the previous work (reference 28) and the new data in the revised manuscript in

The abstract: “The computational data are analyzed and discussed in the context of the experimental results previously obtained by surface plasmon resonance binding assays.”

Page 2: “The aim of this work is to get deeper insights into PCPE-1/GAG interactions using both SPR inhibition assays and in silico techniques to complement the experimental data obtained in the previous [28] and present work.”; “There was a trend towards an increase in inhibition of PCPE-1-HP interaction with the length of HP oligosaccharides from dp2 to dp8, and then from dp14 to full-length HP chains (Figure 1).

Page 3: “Then we applied the in silico approaches we have previously developed to analyze the binding of PCPE-1 to GAGs at the atomic level atomic, and to determine if these interactions were exclusively electrostatic-driven or if other factors modulate the binding strength.”

Page 5: “Therefore, although the binding strength was similar for CS6 and DS, their preferred binding sites were distinct for these two GAGs, which differ only in the epimerization of glucuronic acid. This could potentially explain the results from surface plasmon resonance binding assays, which showed that CS6 did not inhibit PCPE-1 binding to HP whereas DS did [24].

Page 15: “The increase in binding affinity with the increase of HP length agrees with the experimental trend observed in this study (Figure 1).”

Page 20: “CS6 is predicted to bind to a different site of the NTR domain than the other GAGs, which may account for the experimental differences previously observed between CS6 and DS/HP [28].

Round 2

Reviewer 1 Report

accept